# Comparison of Sacral Erector Spinae Plane Block vs. Ring Block for Postoperative Analgesia Management Following Circumcision Surgery: A Prospective, Randomized, Controlled Multicenter Trial

**DOI:** 10.3390/healthcare13060653

**Published:** 2025-03-17

**Authors:** Muhammed Halit Satıcı, Mahmut Sami Tutar, Betül Kozanhan, Yasin Tire, Bülent Hanedan, İlhami Aksoy, İbrahim Akkoyun, Mehmet Emin Boleken, Nuray Altay

**Affiliations:** 1Department of Anesthesiology and Reanimation, Konya City Hospital, University of Health Sciences, 42020 Konya, Turkey; masatu42@gmail.com (M.S.T.); betulkozanhan@gmail.com (B.K.); dryasintire@hotmail.com (Y.T.); bhanedan@hotmail.com (B.H.); ilhamiiaksoy@gmail.com (İ.A.); 2Department of Pediatric Surgery, Konya City Hospital, University of Health Sciences, 42020 Konya, Turkey; ibrahimakkoyun@yahoo.com; 3Department of Pediatric Surgery, Faculty of Medicine, Harran University, 63300 Sanliurfa, Turkey; mboleken@yahoo.com; 4Department of Anesthesiology and Reanimation, Faculty of Medicine, Harran University, 63300 Sanliurfa, Turkey; nurayaltay@ymail.com

**Keywords:** circumcision, FLACC scale, ring block, sacral erector spinae plane block

## Abstract

**Background and Objectives:** Circumcision is the most frequently performed surgery in male pediatric patients. The postoperative period is characterized by significant pain due to the sensitivity of the foreskin and low pain threshold in children. This study aimed to evaluate the effects of sacral erector spinae plane block (S-ESPB) and ring block on postoperative face, legs, activity, cry, and consolability (FLACC) pain scores after circumcision in children. We also assessed the amount of rescue analgesia used, the time to the first administration of rescue analgesia, potential problems, and parental satisfaction. **Materials and Methods:** This study was a prospective, randomized, multicenter trial conducted at two tertiary healthcare centers in Turkey. The patients were divided into two groups: Group S (patients who received the S-ESPB) and Group R (patients who received the ring block). The primary outcome measure was the FLACC score at 1 h postoperatively. Secondary outcome measures included FLACC scores at 0, 2, 4, and 6 h after surgery, the total dose of rescue analgesia, time to first rescue analgesia, complications, and parental satisfaction. **Results:** Group S exhibited significantly lower FLACC scores than Group R at all time (0, 1, 2, 4, and 6 h) points (respectively, *p* = 0.013, *p* < 0.001, *p* = 0.004, *p* = 0.006, and *p* = 0.002). Group S required significantly less rescue analgesia and exhibited a significantly longer duration of analgesic efficacy compared to Group R (*p* = 0.001 and *p* = 0.002, respectively). **Conclusions:** The S-ESPB is a safe and effective form of analgesia for managing pain following pediatric circumcision surgery.

## 1. Introduction

Circumcision is a surgical procedure in which the foreskin covering the glans of the penis is removed. It is the most frequently performed surgery in male pediatric patients [1]. The postoperative period is characterized by significant pain due to the sensitivity of the foreskin and low pain threshold in children. Postoperative pain can cause agitation, delayed wound healing, delayed mobilization, and bleeding from the surgical site [2]. Various analgesic treatments have been developed to alleviate the discomfort. These treatments usually include medical therapy, local anesthetic (LA), neuraxial anesthesia, and peripheral nerve block [3]. Among the nerve blocks, ring block and penile block are routinely applied for analgesia in circumcision surgery [4,5]. A ring block for the penis is administered by injecting LA around the base of the penis in a ring shape [4]. The short duration of the effect of the ring block, its low analgesic power, and its association with complications such as bleeding and hematoma have necessitated the investigation of alternative LA methods [6,7]. The erector spinae plane block (ESPB) is a popular trunk block used in surgical procedures and has been found effective in adult and pediatric patients. Sacral ESPB (S-ESPB) is a novel technique described by Tulgar et al. involving ultrasound-guided identification of the sacral medial crest, followed by lateral movement of the probe to visualize the intermediate sacral crest and injection of LA between the multifidus muscle and the intermediate sacral crest [8]. The S-ESPB is alternatively referred to as the sacral multifidus plane (SMF) block [9]. Various studies have evaluated the efficacy of S-ESPB. The first study of this block demonstrated its application in pilonidal sinus surgery [8]. Since then, it has been used in many other surgeries such as sacral, orthopedic, and pediatric surgery for hypospadias or anoplasty [10,11,12,13,14,15]. The mechanism by which S-ESPB provides analgesia in many surgeries has been debated and remains unclear. Many case reports have hypothesized that LA spreads anteriorly (to the sacral and pudendal nerves, as in gender reassignment surgery) [16] and cephalad towards the lumbar plexus (in orthopedics) [17,18]. This pattern of spread suggests that S-ESPB may provide both somatic and visceral analgesia, particularly through its action on the sacral plexus, pudendal nerve, and lower lumbar nerves. A recent anatomical and radiologic study demonstrated the presence of radiopaque dye in the superficial plane (subcutaneous tissues, S1–S5) and the erector spinae compartments (L2-S3) in both the median and intermediate approaches to the ESPB. Importantly, anterior transition of the radiopaque dye towards the sacral nerve roots (S2–S5) was observed only in tomography scans from the median approach [19]. Another study further elucidated the spread of local anesthetic in SMF block, revealing that dye reached the dorsal sacral foramina upon dissection below the SMF. Axial sections demonstrated that methylene blue dye was observed both above and below the SMF muscle and around the sacral nerve roots. The sagittal view depicted the dye surrounding the SMF muscle and reaching the sacral canal, staining the sacral nerve roots. These findings suggest that the S-ESPB may exert its analgesic effect not only by direct spread to the sacral plexus but also through diffusion into the sacral canal, potentially contributing to extended analgesic coverage [20]. This anterior spread towards the sacral and pudendal nerves suggests that S-ESPB may provide extended analgesia in surgeries involving the perineal and pelvic regions, including circumcision.

This study aimed to compare the effects of S-ESPB and ring block on face, legs, activity, cry, and consolability (FLACC) pain scores, overall amount of rescue analgesia used, time to first administration of rescue analgesia, complications, and parental satisfaction after circumcision among pediatric patients. Our main hypothesis was that the use of S-ESPB following circumcision in pediatric patients would result in significantly lower FLACC pain scores than those from ring block.

## 2. Materials and Methods

### 2.1. Ethics Approval and Registration

This study was approved by the ethics committees of Harran University Faculty of Medicine on 30 October 2023 (approval no. 23.20.33) and of Konya City Hospital on 2 November 2023 (approval no. 11-37). The trial is registered on the ClinicalTrials.gov database (number NCT06136000). This study was conducted in accordance with the 2013 revision of the Declaration of Helsinki. We strictly apply the guidelines outlined in the Consolidated Standards of Reporting Trials (CONSORT) statement [21]. Before randomization, written and verbal consent was obtained from participants, verbal consent only from illiterate patients, and written and verbal consent was obtained from their parents.

### 2.2. Patient Population and Inclusion/Exclusion Criteria

Participants were selected from patients who underwent circumcision surgery at the two participating centers between December 2023 and April 2024. The age range was 4–16 years. The inclusion criteria were receipt of general anesthetic and category I–II for physical status on the American Society of Anesthesiologists (ASA) classification system. The exclusion criteria were lack of parental or patient consent, LA allergies, bleeding problems (in cases of hemophilia, just those due to factor deficiency), presence of infection in the area where the nerve block would be applied, and emergency cases.

### 2.3. Randomization

This was a prospective, randomized, multicenter trial. Randomization into two equally sized groups was performed centrally using a sealed envelope technique. Sequentially numbered opaque envelopes were prepared before the initiation of the study and distributed equally to each participating center. The envelopes were opened only after patient enrollment to ensure allocation concealment and prevent selection bias. Randomization was performed by an anesthesiologist at each center (M.S.T., N.A.) who was not involved in patient follow-up, data collection, or analysis. Regular communication between principal investigators at both centers ensured consistent adherence to the randomization sequence, methodological standardization, and minimized inter-center variability. Randomized participants in Group R received ring blocks and those in Group S received S-ESPB. At both centers, ring blocks were performed by pediatric surgeons with at least 5 years of experience, and S-ESPBs were performed by anesthesiologists with at least 5 years of experience. These individuals did not participate in any other aspects of the study. After the surgery, two different anesthesia specialists, who were blinded to patient allocation, recorded the primary and secondary outcomes of the study. Pain scores and rescue analgesia use were assessed by independent observers who were also blinded to patient allocation. The researchers who intervened, the participants, the analyzer, and the individuals recording the primary and secondary outcomes were blinded to the details of the study and were unaware of the patient group allocations.

### 2.4. Standard General Anesthesia and Postoperative Analgesia Protocol

All patients underwent routine anesthesia monitoring, which included electrocardiogram (EKG), oxygen saturation (SpO_2_), blood pressure (noninvasive), and end-tidal carbon dioxide (EtCO_2_). A 22-gauge intravenous (IV) cannula was inserted, and isotonic fluid was administered at a dosage of 15 mL/kg/h. General anesthesia was induced with 1 mg IV midazolam, 2 mg/kg IV propofol, and 1 µg/kg IV remifentanil. A laryngeal mask airway was inserted, and anesthesia was maintained with 2% sevoflurane in 50% oxygen and 50% air. During surgery, sevoflurane was titrated according to minimum alveolar concentration (MAC). Under general anesthesia, patients received either S-ESPB or a ring block before the surgical procedure began. All patients underwent circumcision surgery using the same surgical method. After circumcision, 15 mg/kg IV paracetamol was administered. Upon transfer to the post-anesthesia care unit, the patients’ FLACC pain scores were postoperatively assessed at hours 0, 1, 2, 4, and 6. Ibuprofen was administered as rescue analgesia to patients who exhibited a FLACC score of 4 or higher. In the first two postoperative hours, ibuprofen was administered rectally at 10 mg/kg. After oral intake was initiated, ibuprofen was administered orally at the same dose (10 mg/kg).

#### 2.4.1. Ultrasound-Guided Sacral Erector Spinae Plane Block

When Group S patients were under general anesthesia and before beginning surgery, they were positioned on the lateral side. Aseptic conditions were ensured for the block, and a linear ultrasound (US) probe (MyLab™ Five; Esaote Europe BV, Maastricht, The Netherlands) was placed on the L5 vertebra spinous process in the sagittal plane to identify the starting point of the sacrum. The probe was moved caudally to visualize the sacral medial crest. Continuing in the lateral direction, the probe was used to locate the intermediate sacral crest. After obtaining an optimal US image at the S2 level, a block needle (50 mm, 21G short bevel) (Stimuplex A, B. Braun, Germany) was advanced craniocaudally until bone contact was achieved. Before LA injection, the needle was aspirated to exclude inadvertent intravascular placement of the tip. Marcaine (0.25%; 0.5 mL/kg; maximum 10 mL) was injected between the multifidus muscle and the intermediate sacral crest. The same block procedure was repeated on the patient’s other side (Figure 1).

#### 2.4.2. Ring Block

Pediatric surgeons routinely perform ring blocks during circumcision at our centers. Under general anesthesia, before initiating the surgical procedure, the pediatric surgeon administered non-US-landmark guided LA. Using a 25-gauge needle, the base of the penile shaft was injected at 3, 6, 9, and 12 o’clock positions with 0.5 mg/kg (maximum 20 mg or maximum 8 mL) of 0.25% marcaine. The LA mixture was administered equally at these clock positions.

### 2.5. Outcome Measures

The primary outcome measure was postoperative FLACC score at 1 h after surgery. The scores range from 0 to 10, with 0 indicating no pain, 1–3 indicating mild pain, 4–6 indicating moderate pain, and 7–10 indicating severe pain [22].

The secondary outcome measures were FLACC scores at 0, 2, 4, and 6 h after surgery, the total dose of rescue analgesia administered to each patient, the time between surgery and the first rescue analgesia, complications, and parental satisfaction. Additionally, the age, weight, height, surgery duration (the surgery time was defined as the duration from the surgical incision to the completion of the surgical procedure), and ASA classification of each patient were documented. Parental satisfaction was assessed using a Likert scale, with 1 indicating “not satisfied at all”, 2 “unsatisfied”, 3 “neutral”, 4 “satisfied”, and 5 “very satisfied”.

### 2.6. Sample Size and Statistical Analyses

The primary outcome of this study was the postoperative FLACC scores 1 h after surgery. A 2-point difference in FLACC scores 1 h after surgery was considered clinically important [23]. A pilot study including both the ring block group (n = 10) and the S-ESPB group (n = 10) was conducted to determine the sample size. In the ring block group, the postoperative FLACC score at 1 h after surgery was 4.1 ± 1.9, whereas it was 2.0 ± 0.82 in the S-ESPB group. Based on these data, using an independent groups *t*-test model with a Cohen’s D effect size of 1.052, it was calculated that each group would need 21 patients to achieve 95% power with a maximum type I error of 5%. Considering the potential dropout rate, the required sample size was determined to be 25 patients per group, resulting in a total of 50 patients. The study was stopped when the specified 50 patients were reached.

The data collected from the study were analyzed using IBM SPSS Statistics for Windows, version 26.0 (IBM Corp., Armonk, NY, USA). The Shapiro–Wilk test was used to assess the normality of data distribution. Continuous variables are expressed as mean and standard deviation or median (25–75 percentiles). Categorical variables are presented as frequencies and percentages. Continuous variables with a normal distribution were analyzed using the independent samples *t*-test, while those without a normal distribution were analyzed using the Mann–Whitney U test. Pearson’s chi-square test was used to compare categorical variables. A Kaplan–Meier curve was constructed to address the first need for rescue analgesia, and groups were compared using the log-rank test. A *p*-value < 0.05 was considered statistically significant within a 95% confidence interval.

## 3. Results

Among the 56 patients found eligible for this study, 6 had their surgeries canceled because of upper respiratory tract infections. The remaining 50 patients were randomized and treated according to the study protocols for the relevant group, with 25 patients in Group R and 25 in Group S (Figure 2).

The demographic characteristics of the patients were comparable between the groups, with no significant differences in age, height, weight, surgery duration, or ASA classification (Table 1).

Postoperative pain intensity in the two randomized groups was evaluated using the FLACC scale. The postoperative pain scores were compared between groups and time points.

Group R exhibited statistically significantly higher FLACC scores at all time (0, 1, 2, 4, and 6 h) points (respectively, *p* = 0.013, *p* < 0.001, *p* = 0.004, *p* = 0.006, and *p* = 0.002) (Table 2).

Comparisons between groups were made for parental satisfaction (measured by a Likert scale ranging from 1 to 5), rescue analgesic consumption (as mg/kg of ibuprofen), time to first analgesic requirement, and postoperative complications during the first 6 h after surgery.

At the end of the study, postoperative rescue analgesia was applied to 88% of all patients (44 patients). When the groups were compared in terms of rescue analgesia application, the time until the first analgesic requirement was shorter in Group R patients than in Group S, and this difference was statistically significant (1 [0–1] vs. 4 [2–6]; *p* = 0.002) (Figure 3).

Additionally, higher doses of rescue analgesics were required in Group R than in Group S at baseline and hours 1, 2, 4, and 6. Significant differences were observed between the groups at only hours 1 and 4 (*p* = 0.004 and *p* = 0.021, respectively) (Figure 4). Group R required more rescue analgesia than Group S (as mg/kg of ibuprofen) (27.3 [20–30] vs. 10 [10–10]; *p* < 0.001). In the final assessment of the secondary outcomes, we found that parental Likert ratings were significantly lower in Group R than in Group S (3 [3–3] vs. 4 [4–5]; *p* = 0.001). Complications were observed in four patients, all in Group R, with three patients having hematomas, and one patient experiencing urinary retention. No complications were observed in Group S. However, a between-group comparison of complications showed that the difference was not statistically significant (*p* = 0.110).

## 4. Discussion

In this prospective, randomized, multicenter study, S-ESPB was more effective than ring blocks for postoperative pain relief after circumcision in pediatric patients. Pain after circumcision is typically severe for the first 2 h, but acute pain and discomfort persist for some time [24]. The ring block was first used in circumcision surgery by Broadman et al. This block is simple to administer, has a short-duration effect, and provides low analgesic power [7]. S-ESPB was first used for analgesic purposes by Tulgar et al. in 2019 in pilonidal sinus surgery [8]. Subsequently, this technique was used for postoperative analgesia after various pediatric procedures, and its usefulness has been well documented [12,13,14,15,17].

The precise mechanism of action of S-ESPB remains unclear; however, recent anatomical and radiological investigations provide insights into its analgesic efficacy. Keleş et al. demonstrated in their cadaveric study that radiopaque dye injected via median and intermediate approaches spread into superficial tissues (S1–S5), as well as erector spinae compartments extending from L2 to S3. Interestingly, the anterior spread of dye towards the sacral nerves (S2–S5) was only observed with the median approach [19]. These anatomical findings support the hypothesis that the analgesic effect of S-ESPB in circumcision surgery may result from the spread of local anesthetic anteriorly towards sacral and pudendal nerve branches. Further anatomical studies and clinical correlation are needed to confirm these findings and fully elucidate the mechanism underlying S-ESPB’s effectiveness in pediatric circumcision analgesia.

S-ESPB has been applied in pediatric surgeries, including hypospadias [12,13,14], abdominal surgery, and lower extremity procedures [17,25]. These studies consistently reported that S-ESPB significantly reduces postoperative pain scores and decreases the need for rescue analgesia. In our study, FLACC scores were consistently lower at all time points in Group S compared to Group R. Patients who received the ring block exhibited higher FLACC scores at baseline and 1, 2, 4, and 6 h post intervention. This difference was most prominent between the baseline and the second hour, suggesting that the ring block was insufficient to manage acute circumcision pain adequately during the first two hours. The low FLACC scores observed in our study demonstrate that S-ESPB provides superior pain control compared to the ring block in pediatric patients. This superiority was particularly evident during the first two postoperative hours when pain intensity tends to peak after circumcision.

In another study, patients who received S-ESPB had significantly lower FLACC scores at the fourth and sixth postoperative hours, underscoring the effectiveness of S-ESPB in controlling pain during the later stages of recovery [26]. However, our findings suggest a more comprehensive and consistent analgesic effect of S-ESPB across all time points, including the early acute phase (0, 1, 2 h). The primary distinction between these studies lies in the ability of S-ESPB to effectively manage acute pain, particularly in the critical first two hours following circumcision. Our results highlight the importance of early postoperative pain control, as circumcision is associated with sharp, acute pain immediately following the procedure. Effective management of this acute pain is crucial for improving patient comfort and reducing the need for early rescue analgesia. Our study not only supports previous research on the efficacy of S-ESPB but also extends these findings by demonstrating its superior efficacy in the critical early postoperative period, thereby reinforcing its potential as a superior pain management technique.

The primary regional anesthesia techniques used for circumcision analgesia include caudal block, penile block, pudendal block, and ring infiltration [27]. Previous studies have demonstrated the superiority of regional techniques over penile ring infiltration in terms of prolonged analgesia and a reduced need for rescue analgesia [4,28]. Among these, caudal block has been shown to provide more effective and longer-lasting analgesia than penile block [29,30]. Caudal block, although highly effective in providing analgesia, carries the risk of motor blockade and urinary retention, making it less ideal for ambulatory procedures [29]. Pudendal block offers broader coverage of the perineal and penile regions but requires more technical expertise and carries a higher risk of vascular injury [31].

Studies have shown that patients who receive S-ESPB experience a reduced need for rescue analgesia during the postoperative period and a longer delay in the time to the first use of rescue analgesia [32,33,34,35]. The duration of ring block after circumcision surgery is short, and patients usually need rescue analgesia early on [4,6]. Although our study was limited to the first 6 h after surgery, the number of patients requiring rescue analgesia and the amount of rescue analgesia consumed were lower in patients who underwent S-ESPB than in patients who underwent ring block. Thus, S-ESPB appears to minimize the need for rescue analgesia by providing pain relief and comfort. Notably, some patients who received S-ESPB did not feel the need for rescue analgesia. This indicates that, in some cases, S-ESPB alone can provide sufficient analgesia, eliminating the need for additional analgesics. Effective analgesia provided by S-ESPB increases patient comfort and reduces analgesic consumption, thereby reducing related side effects and costs. Because the patients were discharged in the eighth hour after surgery, we did not have any information about pain at night or the following day. This is a limitation of our study.

Although ring block is commonly used in circumcision surgery, it is associated with a risk of bleeding and hematoma formation [6]. By contrast, the use of S-ESPB during hypospadias surgery, which is also performed in the penile region, has been found to minimize the risks of these complications [13]. Although S-ESPB is a deeper block, it is associated with a lower risk of complications than ring block because it is away from critical areas such as vascular structures. We observed no complications in patients who received the S-ESPB block. Among those who received ring block, three patients developed hematomas, and one experienced urinary retention. These findings indicate that S-ESPB offers significant advantages in terms of both efficacy and safety, supporting its increased use in relevant pediatric surgeries, especially those involving the urogenital system.

We found that parental satisfaction scores on a Likert scale were significantly higher for patients who received S-ESPB than for those who received ring block. This result demonstrates that S-ESPB increases parents’ confidence in the recovery process of their child. It also implies that the S-ESPB provides more effective pain management than the ring block.

### Limitations

The limitations of this study include the fact that circumcision is a day surgery, and patients are discharged approximately 8 h after the procedure (after confirming that there is no motor weakness or globe development). Although we followed the first 6 h in our study, our main aim was to evaluate acute pain in the first few hours. However, the lack of a long-term follow-up prevented us from assessing the prolonged analgesic effects of S-ESPB beyond this early phase. Since the duration of action of S-ESPB in pediatric patients is not well established, it remains unclear whether its effects extend beyond the first postoperative hours and if it reduces the likelihood of delayed pain during the first night or following days. Additionally, we could not evaluate whether patients experienced breakthrough pain requiring additional analgesics at home. Although post-discharge follow-up via phone is less reliable than face-to-face follow-up, it is preferable to no post-discharge follow-up. The revision of the short-term follow-up duration of 6 h in this study will allow assessments of the long-term sustainability of the analgesic effects provided by S-ESPB. Future studies with longer follow-up durations, including pain assessments at later time points (e.g., 12, 24, and 48 h postoperatively), as well as objective measures of analgesic consumption at home, will help determine the sustained benefits of S-ESPB for postoperative pain management in pediatric patients. A further limitation of our study was that the ring blocks were performed by surgeons rather than anesthesiologists. Additionally, sacral ESP block may hypothetically cause motor weakness. Although no such cases were reported in our study, this potential risk should be considered. Future studies in which objective motor function assessments are performed postoperatively may provide further insights into this theoretical concern.

## 5. Conclusions

The S-ESPB group exhibited significantly lower pain scores and analgesia requirements, and significantly higher parental satisfaction and complication-free analgesia compared with the ring block group. Based on our study findings, S-ESPB can be safely and effectively used for pain control in pediatric patients undergoing circumcision surgery.

## Figures and Tables

**Figure 1 healthcare-13-00653-f001:**
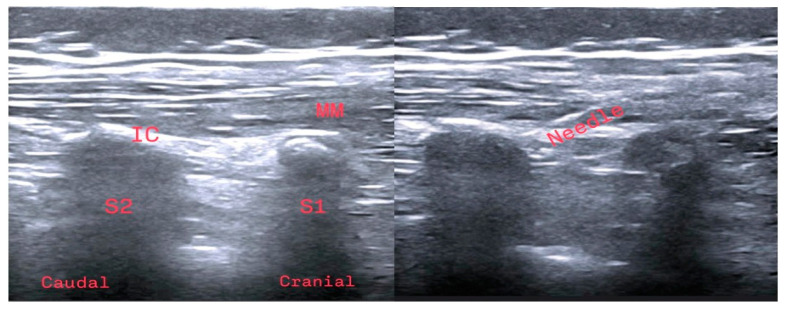
Anatomical representation of the sacral erector spinae plane block under ultrasound guidance. IC: intermediate crest; MM: multifidus muscle; S1: first sacral vertebra; S2: second sacral vertebra. The sacral erector spinae plane block application.

**Figure 2 healthcare-13-00653-f002:**
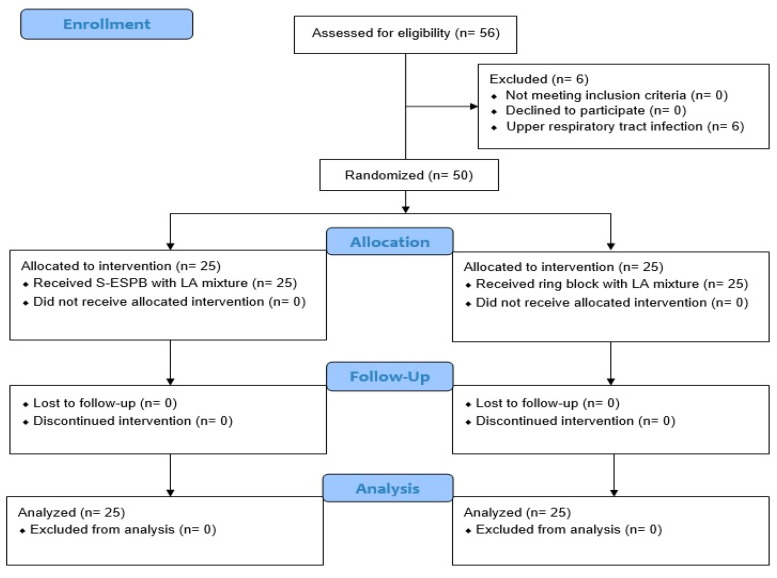
Consolidated standards of reporting trials flow study diagram describing patient progress through the study. S-ESPB, sacral erector spinae plane block; LA, local anesthetic.

**Figure 3 healthcare-13-00653-f003:**
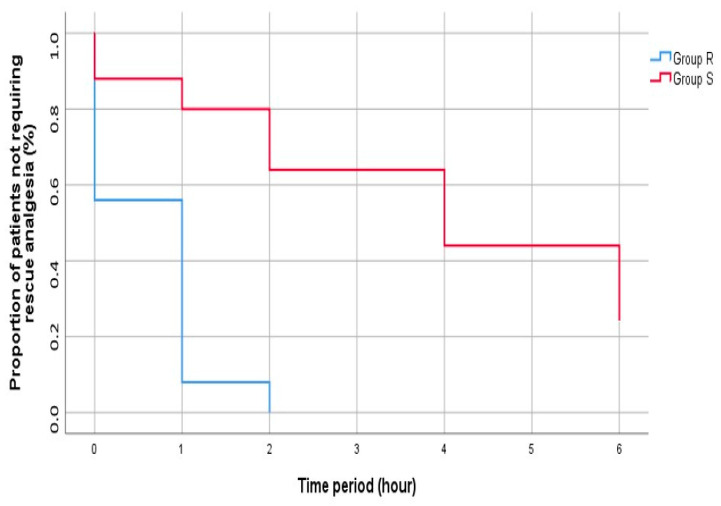
Kaplan–Meier plot showing the percentage of patients not requiring rescue analgesia over time. The Kaplan–Meier curves were compared using the log-rank test.

**Figure 4 healthcare-13-00653-f004:**
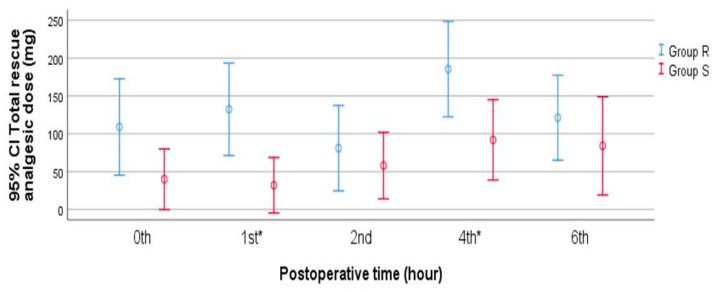
The changes in rescue analgesic doses in different time intervals. *, *p* < 0.05; mg, milligram; CI, confidence interval.

**Table 1 healthcare-13-00653-t001:** Analysis of demographic and clinical variables between groups.

Factors	Group R (n = 25)	Group S (n = 25)
Age (years)	7.56 ± 1.82	8.07 ± 1.88
ASA		
I	23 (92%)	25 (100%)
II	2 (8%)	0 (0%)
Height (cm)	129.76 ± 8.38	129.93 ± 10.46
Weight (kg)	28.24 ± 7.89	31.22 ± 8.40
Surgery time (min)	38.20 ± 11.53	38.15 ± 9.91

Data presented as mean ± standard deviation, or n (%). kg, kilogram; cm, centimeter; min, minutes; ASA, American Society of Anesthesiologists physical status score.

**Table 2 healthcare-13-00653-t002:** Analysis of FLACC scores at different times in groups.

Time (h)	Group R (n = 25)	Group S (n = 25)	*p* Value
0th	3 (3–5)	2 (2–3)	0.013
1st	4 (3–5)	3 (2–3)	<0.001
2nd	4 (3–5)	2 (2–3)	0.004
4th	4 (3–5)	3 (1–4)	0.006
6th	4 (2–5)	2 (0–3)	0.002

Data presented as median (25–75 percentiles). FLACC, face, legs, activity, cry, and consolability.

## Data Availability

The data presented in this study can be made available upon request from the corresponding author due to ethical considerations.

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
