# Peer review of "Comparison of Sacral Erector Spinae Plane Block vs. Ring Block for Postoperative Analgesia Management Following Circumcision Surgery: A Prospective, Randomized, Controlled Multicenter Trial"

_healthcare, 2025, doi:10.3390/healthcare13060653_

Round 1
Reviewer 1 Report
Comments and Suggestions for Authors
Dear authors
I congratulate you for your well-designed work. I have some suggestions for your paper:
- My title suggestion: Comparison of Sacral Erector Spinae Plane Block vs Ring Block for Postoperative Analgesia Management Following Circumcision Surgery: A Prospective, Randomized, Controlled Multicenter Trial
- In the sample size section you specified that the primary outcome of the study was ‘the postoperative FLACC scores 1 hour after surgery’. however in the abstract section you wrote that ‘The primary outcome measures were the 28 assessment of pain at 0, 1, 2, 4, and 6 hours after surgery..’ Please correct the primary outcome in the abstract. Not more than one primary outcome. Specify FLACC scores from other hours as secondary outcomes
- A pediatric sacral ESPB sonoanatomy image will be useful for the paper
- Please correct the primary outcome as the postoperative FLACC scores 1 hour after surgery in the outcome measures seciton.
- In the pilot study for the sample size, there are 10 patients in a ring group. In the pilot study, was there only a ring group or no sacral esp group? For sample size calculation, there should be sacral ESPB group opposite the ring group
- There is no need for subheadings such as primary outcome and secondary outcome in the results
- Discussion and conclusion are well-written
Author Response
Reviewer 1
Comments and Suggestions for Authors
Dear authors
I congratulate you for your well-designed work. I have some suggestions for your paper:
Comment: My title suggestion: Comparison of Sacral Erector Spinae Plane Block vs Ring Block for Postoperative Analgesia Management Following Circumcision Surgery: A Prospective, Randomized, Controlled Multicenter Trial
Response: Thank you very much for your valuable feedback. We have revised the title as per your suggestion. Revised Title:
Comparison of Sacral Erector Spinae Plane Block vs. Ring Block for Postoperative Analgesia Management Following Circumcision Surgery: A Prospective, Randomized, Controlled Multicenter Trial. We appreciate your insightful comments and your time in reviewing our manuscript.
Comment: In the sample size section you specified that the primary outcome of the study was ‘the postoperative FLACC scores 1 hour after surgery’. however in the abstract section you wrote that ‘The primary outcome measures were the 28 assessment of pain at 0, 1, 2, 4, and 6 hours after surgery..’ Please correct the primary outcome in the abstract. Not more than one primary outcome. Specify FLACC scores from other hours as secondary outcomes
Response: Thank you for your valuable feedback. We acknowledge the inconsistency regarding the definition of the primary outcome measure. In accordance with your suggestion, we have revised the abstract section to specify that the FLACC score at 1 hour postoperatively is the primary outcome, while FLACC scores at other time points are now categorized as secondary outcomes. The revised abstract now states:
"The primary outcome measure was the FLACC score at 1 hour postoperatively. Secondary outcome measures included FLACC scores at 0, 2, 4, and 6 hours after surgery, the total dose of rescue analgesia, time to first rescue analgesia, complications, and parental satisfaction."
Comment: A pediatric sacral ESPB sonoanatomy image will be useful for the paper
Response: Thank you for your valuable suggestion. In response to your comment, we have incorporated a sonoanatomy image along with an illustration of the block application as Figure 1 in the revised manuscript. We believe that this addition enhances the clarity of the procedural description and provides a more comprehensive visual representation for the readers. Thank you once again for your valuable feedback and for your time in reviewing our work.
Comment: Please correct the primary outcome as the postoperative FLACC scores 1 hour after surgery in the outcome measures seciton.
Response: Thank you for your valuable feedback. In response to your suggestion, we have revised the Outcome Measures section to clearly define the primary and secondary outcomes. The updated text is as follows:
"The primary outcome measure was the postoperative FLACC score at 1 hour after surgery.
The secondary outcome measures were FLACC scores at 0, 2, 4, and 6 hours after surgery, the total dose of rescue analgesia administered to each patient, the time between surgery and the first rescue analgesia, complications, and parental satisfaction.”
Comment: In the pilot study for the sample size, there are 10 patients in a ring group. In the pilot study, was there only a ring group or no sacral esp group? For sample size calculation, there should be sacral ESPB group opposite the ring group
Response: We sincerely apologize for the confusion caused by our wording in the manuscript. To clarify, a pilot study was conducted with 10 patients in each group, including both the Ring Block group (n=10) and the S-ESPB group (n=10), to determine the sample size. To ensure clarity, we have revised the Sample Size section as follows:"A pilot study including both the Ring Block group (n=10) and the S-ESPB group (n=10) was conducted to determine the sample size. In the Ring Block group, the postoperative FLACC score at 1 hour after surgery was 4.1 ± 1.9, whereas it was 2.0 ± 0.82 in the S-ESPB group." We appreciate your keen attention to detail and your valuable feedback.
Comment: There is no need for subheadings such as primary outcome and secondary outcome in the results
Response: Thank you for your valuable feedback. In accordance with your suggestion, we have removed the subheadings such as Primary Outcome and Secondary Outcome from the Results section to improve the flow and readability of the manuscript.
Comment: Discussion and conclusion are well-written
Response: Thank you for your kind words and positive feedback on the discussion and conclusion sections. Your insights have been invaluable in improving the clarity and quality of our work.
Reviewer 2 Report
Comments and Suggestions for Authors
The study compares the effectiveness of Sacral Erector Spinae Plane Block (S-ESPB) vs. Ring Block for postoperative pain management in pediatric circumcision.
Recommendations
- The literature review should be expanded to provide a broader comparison of regional anesthesia techniques commonly used for circumcision pain relief. Additionally, more details on the mechanistic action of S-ESPB should be included, supported by relevant references.
- It is important to clarify whether the individuals assessing pain scores and rescue analgesia use were blinded to patient allocation. Furthermore, additional details should be provided on how randomization concealment was ensured to minimize potential bias.
- The study's limitations should be addressed, particularly the lack of long-term follow-up, as this could impact the overall assessment of the technique's effectiveness over time.
Author Response
Reviewer 2
Comments and Suggestions for Authors
The study compares the effectiveness of Sacral Erector Spinae Plane Block (S-ESPB) vs. Ring Block for postoperative pain management in pediatric circumcision.
Recommendations
Comment: The literature review should be expanded to provide a broader comparison of regional anesthesia techniques commonly used for circumcision pain relief. Additionally, more details on the mechanistic action of S-ESPB should be included, supported by relevant references.
Response: Thank you for your valuable feedback regarding the need to expand the literature review on regional anesthesia techniques for circumcision analgesia. In response to your suggestion, we have revised this section to provide a more comprehensive comparison of commonly used techniques. The revised text now reads as follows:
"The primary regional anesthesia techniques used for circumcision analgesia include caudal block, penile block, pudendal block, and ring infiltration [27]. Previous studies have demonstrated the superiority of regional techniques over penile ring infiltration in terms of prolonged analgesia and reduced need for rescue analgesia [4, 28]. Among these, caudal block has been shown to provide more effective and longer-lasting analgesia than penile block [29, 30]. Caudal block, although highly effective in providing analgesia, carries the risk of motor blockade and urinary retention, making it less ideal for ambulatory procedures [29]. Pudendal block offers broader coverage of the perineal and penile regions but requires more technical expertise and carries a higher risk of vascular injury [31]."
We appreciate your insightful comments, which have helped improve the clarity and depth of our manuscript.
Thank you for your insightful comments regarding the mechanistic action of S-ESPB. Specifically, we have included additional anatomical and radiologic evidence to further clarify the spread of local anesthetic anteriorly to the sacral and pudendal nerves and its potential diffusion into the sacral canal via the sacral multifidus (SMF) plane. We have incorporated findings from a recent study demonstrating that methylene blue dye reached the dorsal sacral foramina, surrounded the SMF muscle, and extended into the sacral canal, staining the sacral nerve roots. These findings suggest that S-ESPB may exert its analgesic effect not only through direct sacral plexus blockade but also via diffusion into the sacral canal, potentially enhancing its analgesic coverage. This revision provides a more detailed explanation of the local anesthetic spread, its anatomical pathways, and clinical relevance in circumcision surgery. The updated manuscript now includes this revised explanation to better reflect the mechanistic aspects of S-ESPB. Thank you again for your valuable feedback.
Comment: It is important to clarify whether the individuals assessing pain scores and rescue analgesia use were blinded to patient allocation. Furthermore, additional details should be provided on how randomization concealment was ensured to minimize potential bias.
Response: Thank you for your valuable feedback. We appreciate your insightful comments regarding blinding and randomization concealment. To address your concerns, we have clarified these aspects in the revised manuscript. Specifically:
Blinding of Outcome Assessors: We have explicitly stated that the individuals assessing pain scores and recording rescue analgesia use were blinded to patient allocation to minimize potential bias.
Randomization Concealment: We have further elaborated on the randomization process, ensuring that sequentially numbered, opaque envelopes were opened only after patient enrollment to prevent allocation bias.
The revised section now reads as follows:
This was a prospective, randomized, multicenter trial. Randomization into two equally sized groups was performed centrally using a sealed envelope technique. Sequentially numbered opaque envelopes were prepared before the initiation of the study and distributed equally to each participating center. The envelopes were opened only after patient enrollment to ensure allocation concealment and prevent selection bias. Randomization was performed by an anesthesiologist at each center (M.S.T., N.A.), who was not involved in patient follow-up, data collection, or analysis. Regular communication between principal investigators at both centers ensured consistent adherence to the randomization sequence, methodological standardization, and minimized inter-center variability. Randomized participants in Group R received ring blocks and those in Group S received S-ESPB. At both centers, ring blocks were performed by pediatric surgeons with at least 5 years of experience, and S-ESPBs were performed by anesthesiologists with at least 5 years of experience. These individuals did not participate in any other aspects of the study. After the surgery, two different anesthesia specialists, who were blinded to patient allocation, recorded the primary and secondary outcomes of the study. Pain scores and rescue analgesia use were assessed by independent observers who were also blinded to patient allocation. The researchers who intervened, the participants, the analyzer, and the individuals recording the primary and secondary outcomes were blinded to the details of the study and were unaware of the patient group allocations.
Comment: The study's limitations should be addressed, particularly the lack of long-term follow-up, as this could impact the overall assessment of the technique's effectiveness over time.
Response: Thank you for your valuable feedback regarding the study's limitations. We acknowledge the importance of long-term follow-up in assessing the sustained analgesic effects of S-ESPB. In response to your suggestion, we have revised the limitations section to emphasize the potential impact of the lack of long-term follow-up on the overall assessment of the technique’s effectiveness.
In the revised version, we have explicitly stated that while our study focused on acute postoperative pain, the absence of extended follow-up limits our ability to evaluate delayed pain occurrence, nighttime pain, and potential breakthrough pain requiring additional analgesics at home. We also clarified that the duration of action of S-ESPB in pediatric patients is not well established, making it uncertain whether its effects extend beyond the first few postoperative hours.
Additionally, we have suggested that future studies should incorporate longer follow-up periods (e.g., 12, 24, and 48 hours) and objective measures such as analgesic consumption at home to better understand the long-term efficacy of S-ESPB.
Reviewer 3 Report
Comments and Suggestions for Authors
It was a lot of fun for me to evaluate this well-conducted study comparing a very interesting regional anesthesia technique with the conventional technique. But I have identified some problems that need to be fixed.
- The study was two-centered and randomization was done with a sealed envelope technique. This is a bit confusing. Was there a typo? It would have been more appropriate if you used a program.
- central regional anesthetic: this usage is not very appropriate. It would be more appropriate to say neuraxal anesthesia. Although the language is generally good, I recommend that you review your article in terms of English.
- ''Sacral ESPB (S-ESPB) is a new approach that involves the injection of the LA below the multifidus muscle or its aponeurosis in contact with the bony surface, at the intermediate or median sacral crest. The S-ESPB is alternatively referred to as the sacral multifidus plane block '' is the technique described by Tulgar, called sacral multifidus. Please do not confuse the one described by Aksu with the one described by Tulgar and indicate them separately.
- The anesthesiologist who performed the randomization... Please give the first letters of the name and surname of the anesthesiologist who performed the randomization, for example ''The anesthesiologist (SS)''...
- ''Ibuprofen was administered as rescue analgesia to patients who exhibited a FLACC score of 4 or higher'' clearly indicate the dose of ibuprofen and the route of administration.
- ''The probe was moved caudally to visualize the sacral medial 129
crest. Continuing in the lateral direction, the probe was used to locate the intermediate sacral crest..'' As understood from these sentences, you applied the sacral ESP defined by Tulgar. If so, mention this both in the introduction and in the methodology. - ''Konya City Hospital and Harran University Faculty of Medicine'': It is enough to mention the name of your institutions once in the methodology, do not repeat it too often.
- ''The primary outcome of this study was the postoperative FLACC scores 1 hour after surgery''. If this is the primary outcome, there should be consistency throughout the text.
- Age (yr): An abbreviation like yr is not very acceptable. It is more appropriate to write (years).
- Table 1 is offset in shape, you did not need to centralize the row headings. Please edit.
- Surgery time (min): What is the definition of surgery time? Was it written instead of anesthesia time by mistake? Could 38 minutes be the duration of the entire procedure (from induction to removal of LMA)? Please review.
- If you measured FLACC scores at 5 different times, you should perform Bonferroni correction and p<0.001 should be determined for statistical significance.
- Include the results of the Log-Rank test in the explanation of the Kaplan Meshier test.
- Sacral ESP block may hypothetically cause motor weakness. Although not reported, it is useful to keep this possibility in mind. Please include this hypothesis in the discussion and limitation tabs.
- Add a paragraph discussing your 19th reference and other cadaveric studies to the doctrinal tab.
- ''Based on our study findings S-ESPB can be safely and effectively 305
used for pain control in pediatric patients undergoing appropriate surgical procedures in the lower urogenital system. '' Do you think you have obtained enough data to write this as a conclusion sentence in your study? I think it is too bold a statement. Delete this and rewrite it specifically for circumcision. - It seems like you received help from artificial intelligence (LLM) in translating the manuscript. If so, you can mention it in your acknowledgement.
- '... discharged approximately 8 hours after the procedure.' Of course, after confirming that there is no motor weakness and no globe development, right? Please specify this.
Author Response
Reviewer 3
Comments and Suggestions for Authors
It was a lot of fun for me to evaluate this well-conducted study comparing a very interesting regional anesthesia technique with the conventional technique. But I have identified some problems that need to be fixed.
Comment: The study was two-centered and randomization was done with a sealed envelope technique. This is a bit confusing. Was there a typo? It would have been more appropriate if you used a program.
Response: Thank you for this valuable comment. We have clarified our randomization process accordingly and revised the Methods section as follows: "This was a prospective, randomized, multicenter trial. Randomization into two equally sized groups was performed centrally using a sealed envelope technique. Sequentially numbered opaque envelopes were prepared before the initiation of the study and distributed equally to each participating center. The envelopes were opened only after patient enrollment to ensure allocation concealment and prevent selection bias. Randomization was performed by an anesthesiologist at each center (M.S.T., N.A.), who was not involved in patient follow-up, data collection, or analysis. Regular communication between principal investigators at both centers ensured consistent adherence to the randomization sequence, methodological standardization, and minimized inter-center variability." We acknowledge your suggestion regarding electronic randomization and will consider integrating this method into future studies.
Comment: central regional anesthetic: this usage is not very appropriate. It would be more appropriate to say neuraxal anesthesia. Although the language is generally good, I recommend that you review your article in terms of English.
Response: We appreciate the reviewer’s thoughtful suggestion. As recommended, we have revised the terminology from "central regional anesthetic" to "neuraxial anesthesia" throughout the manuscript. Additionally, the manuscript has been carefully reviewed and edited for language accuracy and clarity.
Comment: ''Sacral ESPB (S-ESPB) is a new approach that involves the injection of the LA below the multifidus muscle or its aponeurosis in contact with the bony surface, at the intermediate or median sacral crest. The S-ESPB is alternatively referred to as the sacral multifidus plane block '' is the technique described by Tulgar, called sacral multifidus. Please do not confuse the one described by Aksu with the one described by Tulgar and indicate them separately.
Response: We thank the reviewer for pointing out this potential confusion. We confirm that the technique we utilized is indeed the one originally described by Tulgar et al. To clearly differentiate this technique from that described by Aksu et al., we revised the manuscript accordingly, as follows:
"Sacral ESPB (S-ESPB) is a novel technique described by Tulgar et al., involving ultrasound-guided identification of the sacral medial crest, followed by lateral movement of the probe to visualize the intermediate sacral crest and injection of local anesthetic between the multifidus muscle and the intermediate sacral crest [8]."
Comment: The anesthesiologist who performed the randomization... Please give the first letters of the name and surname of the anesthesiologist who performed the randomization, for example ''The anesthesiologist (SS)''
Response: We thank the reviewer for this valuable comment. As recommended, we have revised the sentence to clearly indicate the anesthesiologists who performed the randomization as follows:
“Randomization was performed by an anesthesiologist at each center (M.S.T., N.A.)”
Comment: ''Ibuprofen was administered as rescue analgesia to patients who exhibited a FLACC score of 4 or higher'' clearly indicate the dose of ibuprofen and the route of administration.
Response: We thank the reviewer for this important suggestion. As recommended, we have revised the sentence to explicitly indicate both the dosage and the route of administration as follows: "Ibuprofen was administered as rescue analgesia to patients who exhibited a FLACC score of 4 or higher. In the first two postoperative hours, ibuprofen was administered rectally at 10 mg/kg. After oral intake was initiated, ibuprofen was administered orally at the same dose (10 mg/kg)."
Comment: ''The probe was moved caudally to visualize the sacral medial 129
crest. Continuing in the lateral direction, the probe was used to locate the intermediate sacral crest..'' As understood from these sentences, you applied the sacral ESP defined by Tulgar. If so, mention this both in the introduction and in the methodology.
Response: We thank the reviewer for highlighting this important oversight. Indeed, we utilized the sacral ESP block technique as described by Tulgar et al. In accordance with your recommendation, we have clearly stated this technique in both the Introduction and Methods sections.
Comment: ''Konya City Hospital and Harran University Faculty of Medicine'': It is enough to mention the name of your institutions once in the methodology, do not repeat it too often.
Response: We thank the reviewer for this valuable suggestion. As recommended, we have revised the manuscript to mention the names of the institutions only once, referring to them subsequently as "the two tertiary healthcare centers" throughout the methodology section.
Comment: ''The primary outcome of this study was the postoperative FLACC scores 1 hour after surgery''. If this is the primary outcome, there should be consistency throughout the text.
Response: We thank the reviewer for highlighting this important issue. As suggested, we have ensured consistency throughout the manuscript, clearly stating and emphasizing that the primary outcome is the postoperative FLACC scores measured at 1 hour after surgery.
Comment: Age (yr): An abbreviation like yr is not very acceptable. It is more appropriate to write (years).
Response: We thank the reviewer for this suggestion. We have revised the manuscript accordingly, replacing all instances of the abbreviation "(yr)" with the full form "(years)".
Comment: Table 1 is offset in shape, you did not need to centralize the row headings. Please edit.
Response: As recommended, we have revised Table 1, aligning the row headings to the left to improve readability and clarity.
Comment: Surgery time (min): What is the definition of surgery time? Was it written instead of anesthesia time by mistake? Could 38 minutes be the duration of the entire procedure (from induction to removal of LMA)? Please review.
Response: The surgery time was defined as the duration from the surgical incision to the completion of the surgical procedure. We have now clearly stated this definition in the outcome measures section of the revised manuscript.
Comment: If you measured FLACC scores at 5 different times, you should perform Bonferroni correction and p<0.001 should be determined for statistical significance.
Response: We appreciate the reviewer’s valuable comment regarding multiple comparisons. However, we would like to clarify that FLACC scores were not compared across multiple time points within groups. Instead, each postoperative time point (0, 1, 2, 4, and 6 hours) was independently analyzed between the two groups (Group S vs. Group R). Therefore, as each comparison was singular and distinct, applying a multiple-testing correction such as Bonferroni was not deemed necessary. We thank the reviewer for allowing us to clarify this point."
Comment: Include the results of the Log-Rank test in the explanation of the Kaplan Meshier test.
Response: Thank you for your valuable suggestion. We have revised Figure 3 and clearly indicated in the figure legend that the Kaplan-Meier curves were compared using the Log-Rank test, as recommended. The revised figure legend is as follows: “Figure 3. Kaplan-Meier plot showing the percentage of patients not requiring rescue analgesia over time. The Kaplan-Meier curves were compared using the Log-Rank test.”
Comment: Sacral ESP block may hypothetically cause motor weakness. Although not reported, it is useful to keep this possibility in mind. Please include this hypothesis in the discussion and limitation tabs.
Response: We appreciate the reviewer’s insightful suggestion. As recommended, we have included this hypothesis in the limitations section of our manuscript. Specifically, we have stated that sacral ESP block may hypothetically cause motor weakness, although no such cases were reported in our study. This potential risk has been acknowledged, and we have highlighted the need for future studies with objective motor function assessments to further investigate this concern.
Comment: Add a paragraph discussing your 19th reference and other cadaveric studies to the doctrinal tab.
Response: We appreciate the reviewer’s valuable suggestion. Specifically, we have added the following paragraph:
"The precise mechanism of action of S-ESPB remains unclear; however, recent anatomical and radiological investigations provide insights into its analgesic efficacy. KeleÅŸ et al. demonstrated in their cadaveric study that radiopaque dye injected via median and intermediate approaches spread into superficial tissues (S1–S5), as well as erector spinae compartments extending from L2 to S3. Interestingly, anterior spread of dye towards the sacral nerves (S2–S5) was only observed with the median approach [19]. These anatomical findings support the hypothesis that the analgesic effect of S-ESPB in circumcision surgery may result from the spread of local anesthetic anteriorly towards sacral and pudendal nerve branches. Further anatomical studies and clinical correlation are needed to confirm these findings and fully elucidate the mechanism underlying S-ESPB’s effectiveness in pediatric circumcision analgesia."
This revision ensures a more comprehensive discussion of the anatomical basis of S-ESPB, aligning with the reviewer’s recommendation.
Comment: ''Based on our study findings S-ESPB can be safely and effectively 305
used for pain control in pediatric patients undergoing appropriate surgical procedures in the lower urogenital system. '' Do you think you have obtained enough data to write this as a conclusion sentence in your study? I think it is too bold a statement. Delete this and rewrite it specifically for circumcision.
Response: We appreciate the reviewer’s insightful comment. As recommended, we have revised the conclusion to specifically focus on circumcision surgery. The revised sentence is as follows: "Based on our study findings, S-ESPB can be safely and effectively used for pain control in pediatric patients undergoing circumcision surgery."
Comment: It seems like you received help from artificial intelligence (LLM) in translating the manuscript. If so, you can mention it in your acknowledgement.
Response: We appreciate the reviewer’s comment. To ensure linguistic clarity and academic precision, we had the manuscript professionally edited by Enago, a well-established academic language editing service. We have a certificate confirming this professional editing process, which is provided below. The Enago certificate is attached below for reference.
Comment: '... discharged approximately 8 hours after the procedure.' Of course, after confirming that there is no motor weakness and no globe development, right? Please specify this.
Response: As recommended, we have revised the sentence to clarify that patients were discharged only after confirming the absence of motor weakness and globe development. The revised sentence is as follows: "The limitations of this study include the fact that circumcision is a day surgery, and patients are discharged approximately 8 hours after the procedure (after confirming that there is no motor weakness or globe development)."
